# Emergent Technologies for the Extraction of Antioxidants from Prickly Pear Peel and Their Antimicrobial Activity

**DOI:** 10.3390/foods10030570

**Published:** 2021-03-09

**Authors:** Elisabete M. C. Alexandre, Marta C. Coelho, Kardelen Ozcan, Carlos A. Pinto, José A. Teixeira, Jorge A. Saraiva, Manuela Pintado

**Affiliations:** 1LAQV-REQUIMTE, Department of Chemistry, Campus Universitário de Santiago, University of Aveiro, 3810-193 Aveiro, Portugal; carlospinto@ua.pt (C.A.P.); jorgesaraiva@ua.pt (J.A.S.); 2CBQF—Centro de Biotecnologia e Química Fina, Laboratório Associado, Escola Superior de Biotecnologia, Universidade Católica Portuguesa, Rua Arquiteto Lobão Vital 172, 4200-374 Porto, Portugal; mccoelho@porto.ucp.pt (M.C.C.); kardelen.oozcan@gmail.com (K.O.); mpintado@porto.ucp.pt (M.P.); 3CEB—Centre of Biological Engineering, University of Minho, 4710-057 Braga, Portugal; jateixeira@deb.uminho.pt

**Keywords:** prickly pear peel, ohmic heating and high-pressure extraction, phenolics, antioxidant activity, antimicrobial activity

## Abstract

Phenolic compounds are important bioactive compounds identified in prickly pear peel that have important antioxidant and antimicrobial properties. However, conventional thermal extraction methods may reduce their bioactivity, and technologies such as high pressure (HP) and ohmic heating (OH) may help preserve them. In this study, both technologies were analyzed, individually and combined (250/500 MPa; 40/70 °C; ethanol concentration 30/70%), and compared with Soxhlet with regard to total phenolics, flavonoids, and carotenoids as well as antioxidant (ABTS, DPPH, ORAC), DNA pro-oxidant, and antimicrobial (inhibition halos, minimum inhibitory concentration (MIC), minimum bactericidal concentration (MBC), growth curves, and viable cells) activities of prickly pear peel extracts. Total phenolics extracted by each technology increased 103% (OH) and 98% (HP) with regard to Soxhlet, but the contents of total flavonoids and carotenoids were similar. Antioxidant activity increased with HP and OH (between 35% and 63%), and OH (70 °C) did not induce DNA degradation. The phenolic compound present in higher amounts was piscidic acid, followed by eucomic acid and citrate. In general, their extraction was significantly favored by HP and OH. Antimicrobial activity against 7 types of bacteria showed effective results only against *S. aureus*, *S. enteritidis*, and *B. cereus*. No synergetic or additive effect was observed for HP/OH.

## 1. Introduction

The edible portion of the prickly pear (*Opuntia ficus-indica* L.) is the pulp, which is usually eaten raw but can also be processed to produce dehydrated derived products and juices, alcoholic beverages, jams, and jellies. The pulp of commercially ripe fruits represents 45–67% of total fruit, the seeds contained in the pulp represent 2–10%, and the pericarp accounts for 33–55% [1]. The nonedible peel is rich in important compounds such as pectins or polyunsaturated fatty acids and different antioxidant compounds but is considered as waste. However, the extraction of antioxidants to be applied to food, cosmetics, and pharmaceuticals make it an attractive resource of bioactive compounds. Prickly pear peel is a very rich source of betalains and phenolic compounds that are usually related to their preventive effect on health issues, acting mainly in an antioxidant, antiviral, anti-inflammatory, and anticarcinogenic manner [2].

Bioactive compounds can be obtained from the peels using traditional extraction methodologies such as Soxhlet. However, these methods require long extraction times and consume large amounts of solvents, the extraction yields and selectivity are low, and the high temperature may volatilize and degrade several compounds [3]. Recently, innovative and green extraction methodologies are being studied, such as high-pressure, pulse electric field, ohmic heating, ultrasound, and supercritical carbon dioxide methods, among others. High pressure (HP) extraction is usually applied between 100 and 600 MPa at low-to-moderate temperatures. In these conditions, HP is able to protect the structure of the compounds since it acts differently on covalent and noncovalent bonds [4]. For example, enzymes, proteins, and lipids are frequently disrupted by HP treatments while smaller molecules, such as vitamins and pigments, remain unchanged. This happens because HP has low or no effect on covalent bonds, but hydrogen and ionic bonds, as well as hydrophobic interactions (noncovalent), are usually destroyed [3]. Thus, HP extraction reduces the resistance of important compounds to the extraction solvent, mainly due to the effect on the tissues, cellular wall, membrane, and organelles [5]. Ohmic heating (OH) is another emerging technology, where alternating electric current is passed directly through a liquid or solid food product, resulting in an instantaneous generation of heat within the product due to the electrical resistance of the food [6]. In particular, this extraction technology can be used to increase the efficiency of solute diffusion throughout the membrane, known as the electro-osmosis effect, improving extractions and resulting in the extraction of high-quality compounds. 

The main goals of this research are to study HP and OH extractions used individually and sequentially combined to increase the extraction yields of total phenolic, flavonoid, and carotenoid compounds and to increase the antioxidant activity of prickly pear peel extracts. The antimicrobial effect against *Bacillus cereus*, *Staphylococcus aureus*, *Listeria monocytogenes*, *Listeria innocua*, *Salmonella enteritidis*, *Pseudomonas aeruginosa*, and *Escherichia coli* was studied in the selected extracts.

## 2. Materials and Methods 

### 2.1. Prickly Pear Peel Storage and Characterization

Ripe fruits from yellow prickly pears were harvested in the south of Portugal, transported, and peeled manually. The peels were then dried at 40 °C up to 7% moisture, grounded, and stored at −20 °C. A generic characterization of dried and ground prickly pear peel was performed with regard to total lipids (the Soxhlet method), total sugars (the Munson and Walker method), humidity (102 °C), and ash (550 °C). The results from these analyses were 0.7 ± 0.00 g/100 g dry weight, 38.75 ± 0.07 g sugar/100 g dry weight, 7.25 ± 0.07 g/100 g dry weight, and 14.25 ± 0.07 g/100 g dry weight, respectively, which are in accordance with results reported by others [7]. 

### 2.2. Extraction Conditions

HP extractions were made in food-grade polyamide/polyester (PA/PE) bags containing 0.6 g of sample and 30 mL of solvent (ethanol 30% and 70%), which were loaded into industrial-scale HP equipment (Model 55, Hyperbaric, Burgos, Spain) with a pressure vessel of 55 L. The extractions were performed under pressures of 0.1, 250, and 500 MPa. Extractions by OH were performed at 25 kHz (a function generator (Agilent 33.220 A, Bayan Lepas, Malaysia; 1 Hz–25 MHz and 1–10 V) connected to an amplifier system (Peavey CS3000, Meridian, MS, USA; 0.3–170 V) was used to control the system), with temperatures at 40 and 70 °C. The solvents used were the same as the ones used for HP extractions (ethanol 30% and 70%). Treatment time was fixed in 15 min for each treatment. Soxhlet extraction was performed for 7 h as the reference methodology, using ethanol 30% (115 °C) and 70% (85 °C). After the extractions, all extracts were centrifuged (15,000 rpm for 10 min at 4 °C), filtered, and stored at −80 °C for the chemical analyses or freeze-dried, re-suspended in water (1 g/mL), and stored at −80 °C for microbiology assays. All quantifications were performed in triplicate.

### 2.3. Phytochemical Assays

#### 2.3.1. Quantification of Total Phenolics, Flavonoids, and Carotenoids

The Folin–Ciocalteau colorimetric method [8] and aluminum chloride colorimetric method [9] were used to quantify total phenolic and flavonoids using gallic acid and quercetin as standards, respectively. These compounds were expressed as mg of standard equivalent per g of dry weight.

Total carotenoids were quantified spectrophotometrically [10], using β-carotene as standard, and the results also were presented as mg of standard equivalent per g of dry weight.

#### 2.3.2. Antioxidant Activity

Total antioxidant activity was determined by ABTS [11], DPPH [12], and ORAC [13] methods using Trolox standard solutions to create the calibration curves. After the percentage of inhibition was calculated, the results were presented as mg of standard equivalent per g of dry weight.

The ORAC method and all methods described in the followed sections were used to analyze only some selected extracts obtained with 30% ethanol (7 conditions) based on previous results (Table 1).

DNA agarose gel electrophoresis [14] was performed for evaluation of the protection of DNA oxidation. The volume of extract used to load the agarose gel was 400 µL.

#### 2.3.3. LC-ESI-UHR-QqTOF-MS Analyses for Individual Compounds 

Individual phenolic compounds were analyzed using an UltiMate 3000 Dionex UHPLC (Thermo Scientific) coupled to an ultrahigh-resolution Qq-time-of-flight (UHR-QqTOF) mass spectrometer with 50,000 full-sensitivity resolution (FSR) (Impact II, Bruker Daltonics, Bremen, Germany). A gradient elution program, at a flow rate of 0.25 mL min^−1^, 35 °C, and an Acclaim RSLC 120 C18 column (100 × 2.1 mm, 2.2 μm) (Dionex), was used to separate the compounds. The mobile phase used was formed by water (A) and acetonitrile (B), both with 0.1% of formic acid, and the linear gradient was 0–7 min: 0–5%B, 7–9 min: 5–95%B, and 9–15 min: 95–5%B. The volume of sample injected was 1 μL.

For MS analysis, a negative ionization mode was used (spectra acquired over a range from *m*/*z* 20 to 1000) and the parameters were as follows: capillary voltage, 4.5 kV; drying gas temperature, 200 °C; drying gas flow, 8.0 L/min; nebulizing gas pressure, 2 bar; collision RF, 300 Vpp; transfer time, 120 μs; prepulse storage, 4 μs. Postacquisition internal mass calibration used sodium formate clusters, with the sodium formate delivered by a syringe pump at the start of each analysis.

### 2.4. Antimicrobial Activity

#### 2.4.1. Selected Extracts and Microorganisms Studied

From the 7 extraction conditions studied, the best were selected (OH 70 °C; 500 MPa; OH 70 °C and 500 MPa and Soxhlet) with regard to individual compound extraction to continue with the microbiological experiments (ethanol concentration fixed at 30%).

In this study, four Gram-positive (*Bacillus cereus* ATCC 2599, *Staphylococcus aureus* ATCC 25923, *Listeria monocytogenes* 13562, and *Listeria innocua* NCTC 11286) and three Gram-negative (*Salmonella enteritidis* ATCC 13076, *Pseudomonas aeruginosa* ATCC 10145, and *Escherichia coli* ATCC 25922) bacteria were used. Muller Hinton broth (MHB; Biokar Diagnostics, Beauvais, France) was used to resuspend the extracts, which were sterilized by filtration using a 0.22-µm filter (Orange Scientific, Braine-l’Alleud, Belgium). The number of replicates used was 3. 

#### 2.4.2. Well Diffusion Assay

These experiments were developed as described by others [15]. The volume of the extract (1 g/mL) used to fill the wells was 40 µl, and the negative and positive controls used were sterile water (negative control) and a lactic acid solution (40% (*v/v*)), respectively. 

#### 2.4.3. Minimum Inhibitory Concentration and Minimum Bactericidal Concentration

Clinical and Laboratory Standards Institute guideline standards M07-A8 (2019) were used to determine the minimum inhibitory concentration (MIC), which was also previously used by Alexandre et al. [15]. The extracts were diluted in different concentrations between 250 and 0.49 mg/mL and the lowest dilution at which microbial growth was prevented, and the initial viability was reduced by at least 99.9% within 24 h and was considered the minimum bactericidal concentration (MBC) [16].

#### 2.4.4. Growth Inhibition Curves and Viable Cell Determination

Extract solutions at MIC, ½ of MIC, ¼ of MIC, and ⅛ of MIC were prepared and inoculated [15]. For positive control, the wells were filled with inoculated MHB, and, for negative control, only MHB was applied.

The content of each well was scraped, and serial dilutions were performed in ringer solution. The drop method was used to determine total viable counts [17]. Positive and negative controls were also analyzed. Viable cells were determined as the log of CFU/mL.

### 2.5. Statistical Analysis

For total and individual compounds, the means were compared by one-way ANOVA. After the verification of homogeneity of covariance, Tukey’s post-hoc tests were performed to determine the significant pair(s). The significance level was established at *p* < 0.05. 

All the results were normalized prior to principal component analysis (PCA) being developed. Factor analysis was made to reduce and explain data variability. The Varimax method was used to produce orthogonal transformations to the reduced factors to better identify the correlations [15].

## 3. Results

### 3.1. Phytochemical Results 

#### 3.1.1. Total Phenolics, Flavonoids, and Carotenoids

Total phenolics were extracted in higher amounts under any OH and/or HP conditions when compared with Soxhlet extraction (8.14 ± 0.21 mg/g DW; Table 1). For individual treatments, the extraction yields increased significantly by 103% for OH (40 °C; ethanol 30%) and 98% for HP (500 MPa; ethanol 30%) when compared with Soxhlet. These results were not statistically different (*p* < 0.05) among themselves or even when compared with the highest extraction yield obtained with combined extractions, which also allowed an increase of 103% (70 °C; 250 MPa; 30% of ethanol). A possible explanation for this is that HP and OH promote the release of phenolic compounds from their intracellular compartments (cell wall lysis) since the entrance of higher quantities of solvent through their membranes can facilitate the extraction of compounds, making them more extractable. Moreover, HP can cause protein denaturation, which may release the phenolic compounds linked to proteins [18]. Comparing both ethanol concentrations, 30%, in general, resulted in higher amounts of total phenolics extracted (more 8%), as well as 70 °C (more 3%) with regard to 40 °C; comparing both technologies, HP led to lower extraction yields, followed by combined extractions (more 2%) and OH (more 4%). 

The highest extraction yields obtained for total flavonoids were 4.08 ± 0.31, 4.54 ± 0.08, 4.52 ± 0.14, and 4.96 ± 0.41 mg/g DW for OH (70 °C, 70%), HP (250 MPa, 70%), combined (40 °C, 250 MPa, 70%), and Soxhlet (70%) extractions, respectively. All these results were similar, and only when OH was applied alone, the total flavonoid extraction yield was significantly lower than the one obtained with Soxhlet. For total carotenoids, a similar behavior was observed. The highest yields obtained were 0.70 ± 0.02, 0.77 ± 0.02, 0.78 ± 0.00, and 0.80 ± 0.00 mg/g DW for OH (40 or 70 °C, 30%), HP (500 MPa, 30%), combined (70 °C, 250 MPa, 30%), and Soxhlet (30%), respectively. The results from OH and/or HP were similar to Soxhlet extractions except for OH used alone, which resulted in a reduction of 13%. In general, 30% ethanol favored the extraction of total phenolics and carotenoids, while for total flavonoids, the best results were obtained for 70%. Regarding the innovative extraction methods, they assure at least similar extraction yields in relation to Soxhlet extraction for total flavonoids and carotenoids and significantly improved the extraction yields of total phenolic compounds. Moreover, the extraction time used for OH and HPE was fixed in 15 min, while Soxhlet extraction was performed for 7 h. 

Similar results were obtained in other works when phenolic compounds were extracted from prickly pear peels by HP [19]. The authors concluded that the extraction of phenolic compounds increases with high-pressure levels and with time under pressure. Regarding total flavonoids and carotenoids, these authors obtained an increase when the extractions were performed under HP, but they compared the results with extractions performed at atmospheric pressure. Coelho et al. [20] use OH to extract bioactive compounds from tomato byproducts. These authors achieved a recovery rate of polyphenols that was 58% higher than that obtained for control, concluding that OH is a successful extraction process that can be used to recover bioactive compounds [20]. Concerning total carotenoids, the extraction yields obtained were also lower with regard to the traditional method [19]. No study was found on the extraction of prickly pear peels by combining OH and HP. However, in our study, neither a synergetic nor an additive effect was observed by sequentially combining OH and HP.

#### 3.1.2. Antioxidant Activity

Using the ABTS method, increments in antioxidant activity of 63%, 18%, and 41% for OH (70 °C; 30%), HP (250 MPa; 30%), and combined (40 °C; 250 MPa; 30%) extractions were obtained, respectively, in relation to Soxhlet. All these results are significantly different from each other, meaning that OH and HP were able to increase total antioxidant activity in the extracts. Regarding DPPH, similar results were found. The augmentation in antioxidant activity was by 19% and 35% for OH (40 °C; 30%) and combined (70 °C; 500 MPa; 30%), respectively, compared with Soxhlet extraction. However, the highest value obtained for HP, when used alone, was not significantly different from Soxhlet. An ethanol concentration of 30% and an extraction temperature of 70 °C were the best conditions to improve antioxidant activity. At this stage, only the most promising extracts were select to analyze antioxidant activity by ORAC (Table 1) and to continue the study (2, 4, 9, 10, 16, 17, and 20). 

The use of OH alone led to the highest quantifications of antioxidant activity (4.49 ± 0.25 and 5.30 ± 0.64 mg/g DW for 40 and 70 °C, respectively). In relation to Soxhlet extraction, OH increased antioxidant activity by 30% and 54% for each temperature. The increase obtained in pressurized extracts was 17% and 9% for 250 and 500 MPa, and combined extractions allowed increases of 8% and 10% for 40 °C/250 MPa and 70 °C/500 MPa, respectively. The increase in antioxidant activity is related mainly to the increase in the extraction of phenolic compounds, which have antioxidant activity. In another study [19], the antioxidant activity of prickly pear peels was affected firstly by the concentration of ethanol used, then by the high-pressure level, and finally by the processing time, but the authors also verified higher antioxidant activity in extracts obtained by HP. When the antioxidant activity of pineapple cubes processed by heating OH and HP was compared, the authors also obtained higher antioxidant activity for samples treated with OH and HP [21], but the treatments were applied separately.

The evaluation of potential pro-oxidant activity revealed that extracts obtained at 70 °C (OH alone and combined with 500 MPa) did not induce DNA degradation since there was little to no increase in fluorescence intensity, hinting that little-to-no interaction between the compounds and the DNA is occurring. A small DNA degradation was observed for the remaining extracts obtained by HP and OH at 40 °C, applied individually. The extract obtained through Soxhlet extraction did not protect or degrade the DNA. Similar results were obtained for blueberry extracts [14]; however, some authors report that antioxidant compounds such as phenolics may act as pro-oxidant agents, interacting with DNA [22]; thus, more studies are required. 

#### 3.1.3. Individual Compounds

By TOF-MS, 18 individual compounds were quantified and 13 individual compounds were identified (Table 2). The most abundant compounds present in all extracts were piscidic acid, eucomic acid, and citrate, representing 40%, 23%, and 23% of all quantified compounds. The remaining compounds were present in amounts below 1%. These results are in good accordance with the literature [23] on prickly pear fruit, where piscidic acid was also the most abundant bioactive compound found in Spanish and Mexican cultivars. 

In general, all the selected conditions allowed us to obtain higher amounts of the majority of compounds when compared with Soxhlet extraction (Table 2). The extracts obtained by HP alone usually presented lower yields when compared with OH or combined extractions. Citrate was better extracted by all technological methods when compared with Soxhlet, and no significant differences were obtained between HP, OH, or combined methods. The extraction yields of individual compounds increased between 645% and 823% regarding Soxhlet extraction. Piscidic and eucomic acids were also better extracted by new technologies, but it was only for OH and combined extractions that significant increases were observed (between 39% and 58% and between 38% and 45%, respectively, for each acid). Significant increases in piscidic acid, obtained in this work (58%), and hydroxybenzoic acid glycoside (between 99% and 120%) concentrations after prickly pear pulp processing at 600 MPa have been reported in the literature [23]. 

### 3.2. Antimicrobial Activity

#### 3.2.1. Well Diffusion Assay

All inhibition halos that were formed around each extract as a consequence of the presence of the target microorganism were measured. However, only some specific conditions allowed us to obtain these inhibition halos. Extracts obtained by Soxhlet did not show inhibition halos against any bacteria. OH 70 °C allowed us to obtain the highest inhibition halos verified for *S. aureus* and *S. enteritidis* (10 and 9.5 mm, respectively), but for the remaining microorganisms, inhibition halos were not observed. For 500 MPa, 3 inhibition halos of 7, 7.5, and 8 mm were observed for *B. cereus*, *S. aureus*, and *S. enteritidis*, respectively, while for the combination of both extraction methods, only one inhibition halo of 7 mm was observed for *S. enteritidis*.

Thus, this first assay to detect an antimicrobial effect of prickly pear peel extracts revealed that *S. aureus* and *S. enteritidis* were more sensitive to extracts than the remaining bacteria, and the extraction temperature seems to have a significant impact on the formation of inhibition halos for *B. cereus* since an inhibition halo was observed only at 500 MPa. Additionally, the linear correlations obtained between the most individual compounds and the inhibition zones and between total carotenoids and the inhibition zones were, on average, higher than 0.8. The inefficiency of Soxhlet extracts to show good results for antimicrobial activity can be related to low extractions of the most important individual compounds, low antioxidant activity, and also the amount of total phenolics present in lower quantities when compared with emerging technologies. The high temperature and long extraction time may also justify these results. 

#### 3.2.2. Minimum Inhibitory Concentration and Minimum Bactericidal Concentration

For *S. aureus* and *S. enteritidis*, the MIC and MBC obtained were 125 and 250 mg/mL, respectively, independent of extraction conditions, while for *B. cereus*, they were 62.5 and 250 mg/mL, respectively, for extracts obtained under pressure (500 MPa). These results are important since the extracts obtained by Soxhlet did not show antimicrobial activity, but these specific extracts (OH 70 °C) were effective against *S. aureus* and *S. enteritidis* and, at 500 MPa, against *B. cereus*. *S. aureus*, and *S. enteritidis*. *B. cereus* is a pathogenic bacteria and an aerobic spore-forming bacterium; it is one of the most thermal-resistant bacteria known and, due to their production of toxins, may cause food poisoning and, consequently, serious health problems [24]. Considering the results obtained, it should be highlighted that prickly pear peel extract obtained at 500 MPa was the only extract able to inhibit the growth of these three pathogenic bacteria. Combined extractions (500 MPa and OH 70 °C) were effective only against *S. enteritidis.*

#### 3.2.3. Growth Inhibition Curves and Viable Cell Determination

Growth curves were performed using MIC, ½ of MIC, ¼ of MIC, and ⅛ of MIC to better understand the effect of the extracts on each bacteria. Figure 1 shows the inhibition curves observed for *S. aureus* (OH 70 °C and 500 MPa), *S. enteritidis* (OH 70 °C, 500 MPa, and the combination of both), and *B. cereus* (500 MPa). The growth of the three bacteria was significantly affected by each extract, at least at MIC and 1/2 MIC. Antimicrobial activity is dependent on the specific bacteria, and polyphenols have been associated with this activity. The position of the OH group in the aromatic ring of polyphenols and the length of the saturated side chain may cause an inhibitory action in bacteria [25]. These hydroxyl groups can degrade bacteria cell membranes, leading to lysis and the release of cellular content. Additionally, metabolic pathways of bacteria may also be destroyed by hydroxyl groups, which may act in the active site of enzymes [25]. 

Figure 2 shows the microbiological loads for *B. cereus*, *S. aureus*, and *S. enteritidis* for the different extract concentrations analyzed, for an incubation time of 24 h, and for the extractions performed at OH 70 °C, 500 MPa, and the combination of both. For *B. cereus* and *S. aureus*, counts at MIC concentrations were higher than the initial load of each microorganism added to the extracts since these loads were not visually detected during observation. The method used in our study to find all MIC was visual observation, which is not as accurate as the spectrophotometric method. Sometimes, visual observation does not perceive the turbidity caused by cellular growth when microbial loads are low, but these can be detected spectrophotometrically. For the highest concentrations tested (250 mg/mL), all samples led to significant reductions, leading to bacterial death (MBC), with only the HP extract causing bacterial reductions for the three bacteria.

#### 3.2.4. Principal Component Analysis

The three principal components are presented in Figure 3. Conjointly, these components are responsible for 92% of variations. PC 1 is the main component since it explains the highest variation, representing 63% of the total variance. This variation can be mainly attributed to total carotenoids and all individual compounds, which are positively related, and inhibition halos. PC2 explained 18% of the total variance and is mostly associated with antioxidant activity measured by ABTS, DPPH, and ORAC methods and total phenolics. PC3 explained more than 11% of the total variance, and MIC and total carotenoids, which are positively related, were the variables that better describe this variation. In general, all variables were somehow related since antioxidant activity is directly associated with total and individual compounds.

## 4. Conclusions

In general, both emerging technologies (ohmic heating (OH) and high pressure (HP)) allowed higher extraction yields than Soxhlet extraction, higher antioxidant activity, and, in some specific conditions, showed the capability to inactivate some bacteria. For total phenolics extraction, the extraction yields increased by 103%, 98%, and 103% in relation to Soxhlet when the extracts were obtained by OH, HP, or combined, respectively. An ethanol concentration of 30%, 70 °C, and OH used alone obtained the best yields compared to ethanol 70%, 40 °C, and HP extraction. As expected, total antioxidant activity measured by all methods followed a similar trend. Antioxidant activity increased by 63%, 18%, and 41% for OH, HP, and combined extractions, respectively, in relation to Soxhlet, and an ethanol concentration of 30% and 70 °C were also preferable to improve antioxidant activity, which is strongly related to the extractability of phenolic compounds. The evaluation of potential pro-oxidant activity revealed that extracts obtained using 70% ethanol, OH 70 °C, or combined with 500 MPa did not induce DNA degradation, meaning that little or no interaction between the compounds and the DNA occurs. The extraction yields obtained for total flavonoids and carotenoids, in general, were similar to Soxhlet extraction, but an ethanol concentration of 30% for the extraction of total carotenoids is preferred. The most abundant compounds present in all extracts were piscidic acid, eucomic acid, and citrate, representing 40%, 23%, and 23% of all quantified compounds, and, in general, all conditions conduced to the extraction of higher quantities of the majority of compounds when compared with Soxhlet extraction. Among technologies, OH and combined extractions also allowed us to obtain higher extraction yields.

Regarding antimicrobial activity, OH 70 °C allowed us to obtain the highest inhibition halos verified for *S. aureus* and *S. enteritidis*, while 500 MPa was effective against *B. cereus*, *S. aureus*, and *S. enteritidis*; the extract obtained by the combination of both methods inactivated only *S. enteritidis*. All results were important, but the inactivation of *B. cereus* (a spore-forming and highly thermally resistant bacterium) by the extract obtained by high pressure alone (500 MPa) should be highlighted. Moreover, the highest amount of piscidic acid was obtained for the extract obtained with OH 70 °C, which might be related to the highest antioxidant activity quantified in the same extract; this, in turn, may be related to the highest inhibition halos verified for *S. aureus* and *S. enteritidis*. However, more conditions and analyses should be carried out to clearly conclude the relationships between OH 70 °C, piscidic acid, and antioxidant or antimicrobial activity. 

Although no synergetic effect was observed in the combined methods, OH and HP extractions were able to significantly increase total phenolic extraction yields and total antioxidant activity. Moreover, some of these extracts were able to inhibit some specific bacteria. Thus, both technologies have the potential to be applied to increase the success of extractions. 

## Figures and Tables

**Figure 1 foods-10-00570-f001:**
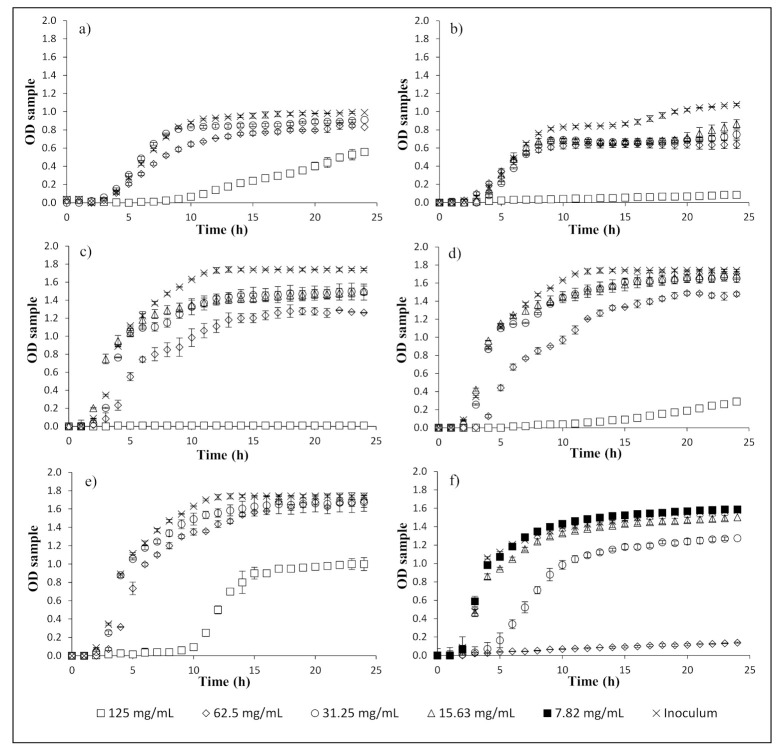
Inactivation curves for (**a**) *S. aureus*/OH; (**b**) *S. aureus*/HP; (**c**) *S. enteritidis*/OH; (**d**) *S. enteritidis*/HP; (**e**) *S. enteritidis*/OH and HP; (**f**) *B. cereus*/HP.

**Figure 2 foods-10-00570-f002:**
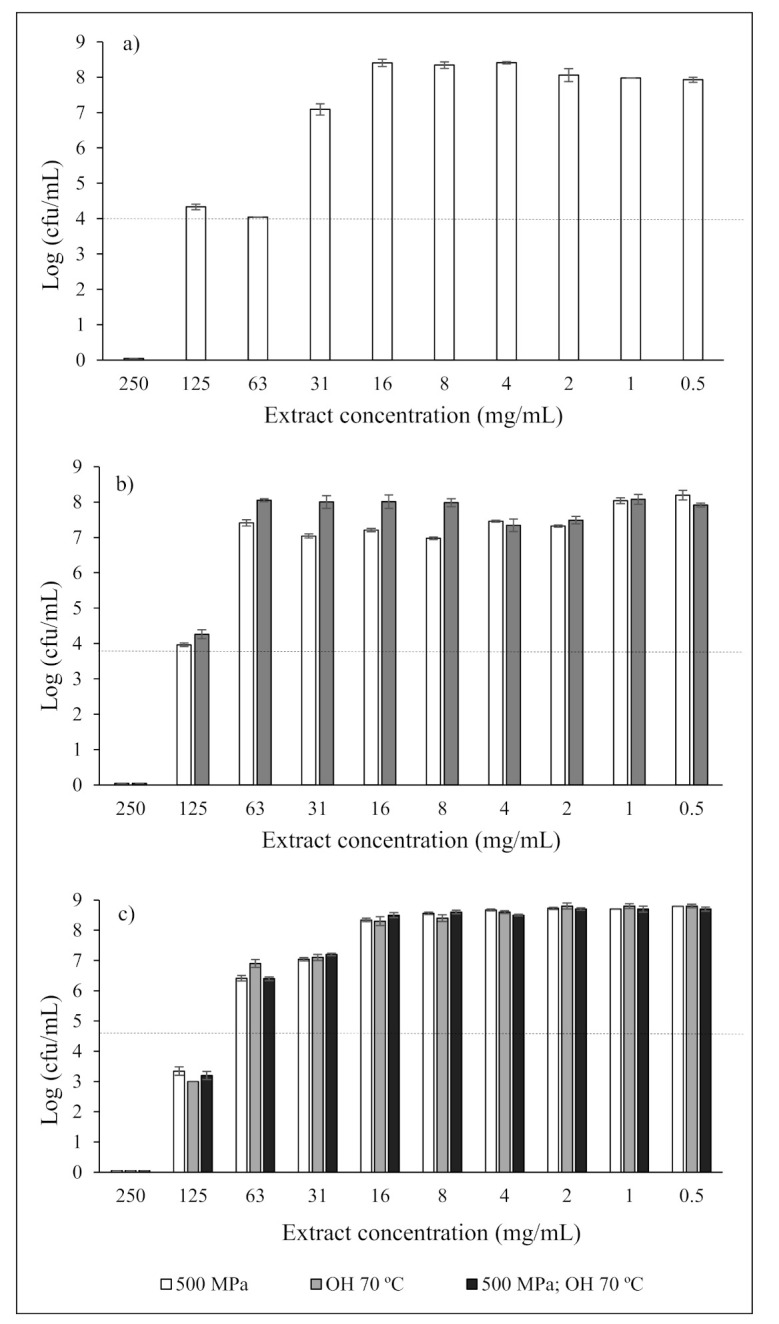
Microbiological loads of (**a**) *B. cereus*, (**b**) *S. aureus*, and (**c**) *S. enteritidis* for the different extract concentrations. The horizontal line in the graphs represents the initial load of microorganism added to the extracts.

**Figure 3 foods-10-00570-f003:**
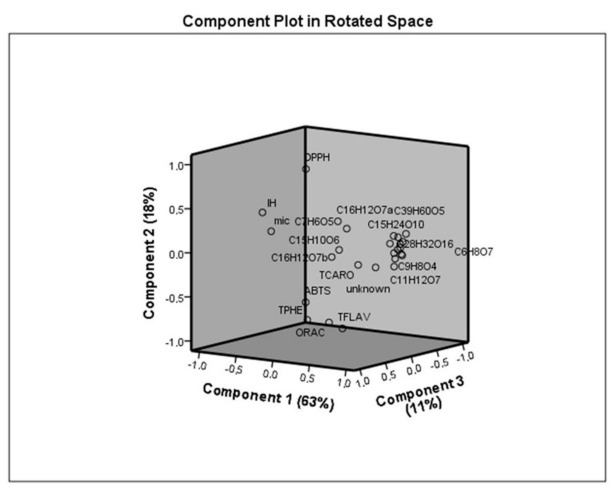
Principal component analysis (PCA) of data, where ABTS, DPPH, ORAC, TFLAV, TPHE, TCARO, IH, and MIC mean total antioxidant activity measured by ABTS, DPPH, and ORAC methods, total flavonoids, total phenolics, total carotenoids, inhibition halos, and minimum inhibitory concentration, respectively. Individual compounds are identified by their chemical formula.

**Table 1 foods-10-00570-t001:** Antioxidant activity, total phenolics, total flavonoids, and total carotenoids of extracts obtained under different extraction conditions.

Samples		Ethanol Concentration (%)	Temperature (°C)	Pressure (MPa)	Total Phenolics (mg/g DW)	Total Flavonoids (mg/g DW)	Total Carotenoids (mg/g DW)	Total Antioxidant Activity (mg/g DW)		
								ABTS	DPPH	ORAC
1	Ohmic heating	70	70	-	15.31 ± 0.34 ^e–i^	4.08 ± 0.31 ^c–f^	0.69 ± 0.04 ^e, f^	7.13 ± 0.17 ^h, i^	4.64 ± 0.18 ^f^	-
2		30	70	-	16.40 ± 0.45 ^i, j^	3.88 ± 0.28 ^b–e^	0.70 ± 0.00 ^f, g^	8.25 ± 0.15 ^j^	5.32 ± 0.13 ^g, h^	5.30 ± 0.64 ^c^
3		70	40	-	14.17 ± 0.63 ^c–e^	3.66 ± 0.28 ^b, c^	0.66 ± 0.00 ^d, e^	6.81 ± 0.12 ^g–i^	3.96 ± 0.07 ^c–e^	-
4		30	40	-	16.53 ± 0.39 ^j^	3.72 ± 0.36 ^b–d^	0.70 ± 0.02 ^e–g^	7.38 ± 0.25 ^i^	5.48 ± 0.12 ^h, i^	4.49 ± 0.25 ^b, c^
5		70	-	0.1	13.47 ± 0.70 ^c^	3.59 ± 0.26 ^b, c^	0.57 ± 0.01 ^a, b^	5.71 ± 0.42 ^c–e^	3.07 ± 0.14 ^a, b^	-
6	High pressure	70	-	250	14.05 ± 0.64 ^c, d^	4.54 ± 0.08 ^f, g^	0.58 ± 0.00 ^b, c^	5.44 ± 0.41 ^b–d^	3.82 ± 0.08 ^c, d^	-
7		70	-	500	14.44 ± 0.53 ^d–f^	4.03 ± 0.21 ^c–f^	0.62 ± 0.00 ^c, d^	5.43 ± 0.39 ^b–d^	3.62 ± 0.48 ^b, c^	-
8		30	-	0.1	13.50 ± 0.21 ^c^	2.39 ± 0.36 ^a^	0.53 ± 0.00 ^a^	4.87 ± 0.24 ^b^	4.24 ± 0.20 ^c–f^	-
9		30	-	250	15.42 ± 0.42 ^f–j^	3.64 ± 0.26 ^b, c^	0.72 ± 0.03 ^f–h^	5.95 ± 0.33 ^d–f^	4.54 ± 0.13 ^e, f^	4.04 ± 0.63 ^a, b^
10		30	-	500	16.11 ± 0.36 ^h–j^	3.34 ± 0.08 ^b^	0.77 ± 0.02 ^i, j^	5.67 ± 0.06 ^c–e^	4.29 ± 0.12 ^d–f^	3.739 ± 0.07 ^a, b^
11	Ohmic heating plus high pressure	70	70	250	14.55 ± 0.62 ^c–f^	4.30 ± 0.22 ^e, f^	0.59 ± 0.02 ^b, c^	5.80 ± 0.43 ^d–f^	4.73 ± 0.27 ^f, g^	-
12		70	70	500	14.93 ± 0.85 ^d–g^	3.98 ± 0.25 ^c–f^	0.63 ± 0.03 ^c, d^	6.68 ± 0.11 ^g–i^	4.62 ± 0.11 ^f^	-
13		70	40	250	15.00 ± 0.68 ^d–h^	4.52 ± 0.14 ^f, g^	0.62 ± 0.00 ^c, d^	5.79 ± 0.31 ^d–f^	2.83 ± 0.05 ^a^	-
14		70	40	500	14.74 ± 0.62 ^d–f^	4.22 ± 0.12 ^d–f^	0.62 ± 0.01 ^c, d^	5.47 ± 0.19 ^b–d^	2.62 ± 0.21 ^a^	-
15		30	70	250	16.50 ± 0.58 ^j^	3.69 ± 0.19 ^b–d^	0.78 ± 0.00 ^i, j^	6.28 ± 0.28 ^e–g^	5.59 ± 0.53 ^h, i^	-
16		30	70	500	16.05 ± 0.35 ^g–j^	3.62 ± 0.24 ^b, c^	0.75 ± 0.01 ^g–i^	6.45 ± 0.34 ^f–h^	6.22 ± 0.26 ^j^	3.785 ± 0.07 ^a, b^
17		30	40	250	15.58 ± 0.34 ^f–j^	3.56 ± 0.24 ^b, c^	0.74 ± 0.04 ^g–i^	7.13 ± 0.12 ^h, i^	5.78 ± 0.31 ^h–j^	3.711 ± 0.05 ^a, b^
18		30	40	500	15.12 ± 0.53 ^d–h^	3.40 ± 0.09 ^b^	0.76 ± 0.01 ^h–j^	6.24 ± 0.15 ^e–g^	5.99 ± 0.34 ^i, j^	-
19	Soxhlet	70	85	-	6.20 ± 0.34 ^a^	4.96 ± 0.41 ^g^	0.71 ± 0.00 ^f, g^	3.92 ± 0.46 ^a^	4.54 ± 0.18 ^e, f^	-
20		30	115	-	8.14 ± 0.21 ^b^	4.87 ± 0.46 ^g^	0.80 ± 0.00 ^j^	5.05 ± 0.63 ^b, c^	4.61 ± 0.03 ^f^	3.444 ± 0.22 ^a^

Values in the same column with the same letters are not significantly different (*p* > 0.05).

**Table 2 foods-10-00570-t002:** Individual phenolic compounds of selected extracts obtained using an ethanol concentration of 30% (mg/g DW).

Equation	Name	[M-H]-	MS/MS	Retention Time (min)	250 MPa	500 MPa	OH 40 °C	OH 70 °C	500 MPa + OH 70 °C	250 MPa + OH 40 °C	Soxhlet
-	-	215.0322	101/71	1.3	0.740 ± 0.024 ^a^	0.710 ± 0.079 ^a^	0.789 ± 0.021 ^a^	0.808 ± 0.017 ^a^	0.800 ± 0.002 ^a^	0.787 ± 0.006 ^a^	0.741 ± 0.078 ^a^
C_6_H_8_O_7_	Citrate	191.0194	160/103/85	1.5	1.994 ± 0.137 ^b^	1.819 ± 0.453 ^b^	2.251 ± 0.085 ^b^	2.248 ± 0.212 ^b^	2.245 ± 0.064 ^b^	2.164 ± 0.002 ^b^	0.244 ± 0.016 ^a^
C_11_H_12_O_7_	Piscidic acid	255.0508	193/179/165	3.0	3.133 ± 0.224 ^abc^	2.971 ± 0.522 ^a, b^	3.499 ± 0.075 ^b–d^	3.899 ± 0.085 ^d^	3.743 ± 0.012 ^c, d^	3.441 ± 0.043 ^b–d^	2.465 ± 0.305 ^a^
C_7_H_6_O_5_	Gallic acid	169.0142	125	3.2	0.005 ± 0.001 ^b–d^	0.004 ± 0.001 ^a, b^	0.004 ± 0.000 ^a–c^	0.006 ± 0.001 ^d^	0.006 ± 0.000 ^c, d^	0.005 ± 0.000 ^b–d^	0.003 ± 0.001 ^a^
C_39_H_6__0_O_5_	-	959.3257	193/175/161	3.2	0.017 ± 0.002 ^a, b^	0.015 ± 0.004 ^a^	0.019 ± 0.001 ^a, b^	0.017 ± 0.001 ^a, b^	0.019 ± 0.000 ^a, b^	0.021 ± 0.000 ^b^	0.016 ± 0.002 ^a, b^
C_15_H_24_O_10_	-	365.1364	350/203/159	3.2	0.004 ± 0.000	0.004 ± 0.001	0.005 ± 0.0000	0.004 ± 0.000	0.003 ± 0.001	0.005 ± 0.000	0.003 ± 0.001
C_9_H_8_O_4_	Caffeic acid	179.0362	135	3.6	0.008 ± 0.000 ^a–c^	0.007 ± 0.002 ^a, b^	0.009 ± 0.0000 ^b–d^	0.010 ± 0.000 ^d^	0.010 ± 0.001 ^c, d^	0.010 ± 0.000 ^c, d^	0.007 ± 0.001 ^a^
C_21_H_30_O_13_	-	489.1613	419/235/193	3.6	0.003 ± 0.000 ^a, b^	0.002 ± 0.000 ^a^	0.003 ± 0.000 ^a, b^	0.003 ± 0.000 ^b^	0.003 ± 0.000 ^a, b^	0.003 ± 0.000 ^a, b^	0.003 ± 0.000 ^a, b^
C_11_H_12_O_6_	Eucomic acid	239.0543	179/149/107	3.6	1.900 ± 0.106 ^a–c^	1.576 ± 0.412 ^a, b^	2.075 ± 0.010 ^b, c^	2.162 ± 0.056 ^c^	2.120 ± 0.014 ^c^	2.099 ± 0.053 ^c^	1.498 ± 0.222 ^a^
C_16_H_20_O_9_	1-Feruloyl-d-glucose	355.1039	193/160	3.7	0.022 ± 0.003 ^a, b^	0.019 ± 0.006 ^a, b^	0.024 ± 0.000 ^b^	0.025 ± 0.001 ^b^	0.024 ± 0.000 ^b^	0.024 ± 0.000 ^b^	0.014 ± 0.001 ^a^
C_34_H_42_O_20_	Glycosylated isorhamnetin	769.2186	299/314/178	3.8	0.056 ± 0.005 ^a–c^	0.048 ± 0.013 ^a, b^	0.064 ± 0.001 ^b, c^	0.067 ± 0.001 ^c^	0.065 ± 0.001 ^b, c^	0.065 ± 0.001 ^b, c^	0.046 ± 0.008 ^a^
C_33_H_40_O_20_	Glycosylated isorhamnetin	755.2044	299/314/178	3.8	0.060 ± 0.005 ^a–c^	0.051 ± 0.015 ^a, b^	0.067 ± 0.001 ^a–c^	0.070 ± 0.000 ^c^	0.069 ± 0.001 ^b, c^	0.070 ± 0.001 ^b, c^	0.050 ± 0.007 ^a^
C_27_H_30_O_16_	Glycosylated isorhamnetin	609.1467	299/314/178	4.0	0.048 ± 0.005 ^a–c^	0.042 ± 0.011 ^a^	0.056 ± 0.001 ^a–c^	0.061 ± 0.002 ^c^	0.059 ± 0.001 ^b, c^	0.059 ± 0.003 ^b, c^	0.045 ± 0.006 ^a, b^
C_28_H_32_O_16_	Glycosylated isorhamnetin	623.1627	299/315/271	4.1	0.071 ± 0.007 ^a, b^	0.057 ± 0.018 ^a^	0.079 ± 0.002 ^a, b^	0.086 ± 0.001 ^b^	0.083 ± 0.004 ^b^	0.077 ± 0.003 ^a, b^	0.063 ± 0.012 ^a, b^
C_21_H_20_O_10_	-	431.0990	193/165/134	4.8	0.019 ± 0.002 ^b, c^	0.015 ± 0.005 ^b^	0.022 ± 0.000 ^c^	0.023 ± 0.001 ^c^	0.023 ± 0.001 ^c^	0.024 ± 0.001 ^c^	0.002 ± 0.000 ^a^
C_16_H1_2_O_7_	3-*O*-Methyl-quercetin	315.0514	300/271/255	5.3	0.008 ± 0.001 ^b, c^	0.005 ± 0.002 ^a^	0.007 ± 0.000 ^a, b^	0.011 ± 0.000 ^d^	0.010 ± 0.000 ^c, d^	0.008 ± 0.000 ^b–d^	0.009 ± 0.002 ^b–d^
C_15_H_10_O_6_	Kampferol	285.0410	185/93	5.6	0.002 ± 0.001 ^a^	0.001 ± 0.000 ^a^	0.002 ± 0.000 ^a^	0.003 ± 0.000 ^a^	0.002 ± 0.000 ^a^	0.002 ± 0.000 ^a^	0.008 ± 0.002 ^b^
C_16_H_12_O_7_	Isorhamnetin	315.0520	311/183/119	5.7	0.002 ± 0.000 ^a^	0.001 ± 0.000 ^a^	0.002 ± 0.000 ^a^	0.006 ± 0.003 ^a^	0.003 ± 0.000 ^a^	0.002 ± 0.000 ^a^	0.069 ± 0.016 ^b^

Values in the same column with the same letters are not significantly different (*p* > 0.05).

## Data Availability

Data are not available.

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
