# Peer review of "Emergent Technologies for the Extraction of Antioxidants from Prickly Pear Peel and Their Antimicrobial Activity"

_foods, 2021, doi:10.3390/foods10030570_

Round 1
Reviewer 1 Report
The present manuscript is dealing with the extraction yields of phenolic compounds from prickly pear peel by high pressure and ohmic heating. The manuscript also investigated the antioxidant and antimicrobial potential of the obtained extracts.
The topic is interesting. Whether the first experimental part (extraction conditions) is well designed and performed, in my opinion the second experimental part (antioxidant activity and antimicrobial activity) is poor of novelty. The greater antioxidant and antimicrobial activity linked to the greater concentration of phenolic compounds is a fairly predictable observation. Thus, in my opinion, the manuscript does not provide a significant advance of literature. For instance, a greater novelty could be provided if the antioxidant or antimicrobial activity were attributed to a specific phenolic compound highly extracted with a specific extraction condition.
The English language and the writing style are fine.
Author Response
The present manuscript is dealing with the extraction yields of phenolic compounds from prickly pear peel by high pressure and ohmic heating. The manuscript also investigated the antioxidant and antimicrobial potential of the obtained extracts.
The topic is interesting. Whether the first experimental part (extraction conditions) is well designed and performed, in my opinion the second experimental part (antioxidant activity and antimicrobial activity) is poor of novelty. The greater antioxidant and antimicrobial activity linked to the greater concentration of phenolic compounds is a fairly predictable observation. Thus, in my opinion, the manuscript does not provide a significant advance of literature.
Answer: Thank you for your pertinent comments! The novelty of this paper is mainly related with the use of two emergent technologies, namely high pressure and ohmic heating, combined or no, as extraction methods to recover bioactive compounds from a by-product. High pressure is a non-thermal method while ohmic heating is thermal and the combination of both could increase the extraction of each compound or even increase the extraction of different compounds. As far as we know, this was the first time that high pressure and ohmic heating were combined as extraction methods to valorise by-products.
For instance, a greater novelty could be provided if the antioxidant or antimicrobial activity were attributed to a specific phenolic compound highly extracted with a specific extraction condition.
The English language and the writing style are fine.
Answer: Our results do not allow concluding clearly that antioxidant or antimicrobial activity obtained can be attributed to a specific phenolic compound that was highly extracted with a specific extraction condition. However, the following sentence was added to conclusions: “Moreover, the highest amount of piscidic acid was obtained for the extract obtained with OH 70 ºC that might be related with the highest antioxidant activity quantified in same extract, which in turn may be related with the highest inhibition halos verified for S. aureus and S. enteritidis. However, more conditions and analyses should be carried out to conclude clearly the relation between OH 70 ºC, piscidic acid and antioxidant or antimicrobial activity.” Please see the attachment.

Reviewer 2 Report
All in all the manuscript is written very well. The introduction gives a good overview of the topic and the applied methods are well described.
However, the manuscript would benefit significantly from more graphical approach in presenting the extraction results presented in Table 1. Data presented in a table are often hard to comprehend for the reader and figures make larger amount of data more accessible and easier to compare for the readers.
Author Response
All in all the manuscript is written very well. The introduction gives a good overview of the topic and the applied methods are well described.
However, the manuscript would benefit significantly from more graphical approach in presenting the extraction results presented in Table 1. Data presented in a table are often hard to comprehend for the reader and figures make larger amount of data more accessible and easier to compare for the readers.
Answer: In general, the authors agree with the reviewer. However, the results of this specific Table are associated to 20 different combined treatments that are impossible to identify in the graphic (only using numbers). But if we use numbers, the readers will not know what combination refers each column and will be more difficult identify each treatment. Moreover, it will be very hard to include the statistic results in the bars because some treatments have 4 different letters and this action will originate 6 long graphics defaulting the comparison of each variable. For all that reasons, we believed that this single Table gives to readers more information allowing a better comparison of the results than separated graphs.
Reviewer 3 Report
In this manuscript Alexandre and co-workers report on their study evaluating the impact of using high pressure and ohmic heating techniques to the extraction of phenolic compounds from prickly pear peel. The approach appears to be sound and relevant to scientists working in this field. The data is presented clearly and supports the conclusions. The authors also include some data on antioxidant activities of the extracts, however, as these do not seem to present any new knowledge, this reviewer feels that these could be omitted in favor of a more concise paper. My main concern is the modest command of grammar and sentence structure that makes reading difficult at times. More effort is needed by the authors, and consultation with a native speaker (or editing service) is recommended.
For instance, line 49; use 'ultrasound' in singular; line 50: better: 'low- to-moderate temperature'; change 'compounds structure' to 'structure of the compounds'; line 51: 'high pressure acts selectively since may disrupt' rephrase and add 'it'...and so on.
Author Response
In this manuscript Alexandre and co-workers report on their study evaluating the impact of using high pressure and ohmic heating techniques to the extraction of phenolic compounds from prickly pear peel. The approach appears to be sound and relevant to scientists working in this field. The data is presented clearly and supports the conclusions. The authors also include some data on antioxidant activities of the extracts, however, as these do not seem to present any new knowledge, this reviewer feels that these could be omitted in favor of a more concise paper.
Answer: Thank you for your pertinent comments! Total antioxidant activity was analysed in 20 different extracts by ABTS and DPPH and in 7 conditions by ORAC. We believe that this is an important and relevant part of our study since we have selected the treatments based on total polyphenol compounds extraction, but also based on total antioxidant activities. The determination of antioxidant activity may reinforce the value-added of the extracts with highest polyphenol contents and also establish a correlation of polyphenol contents with antioxidant activity and relates also with antimicrobial effect. Additionally, the readers may compare antioxidant activity of combined treatments with Soxhlet that is part of the novelty of this study.
My main concern is the modest command of grammar and sentence structure that makes reading difficult at times. More effort is needed by the authors, and consultation with a native speaker (or editing service) is recommended.
Answer: The authors revised all the manuscript and believed that the English is now improved.
For instance, line 49; use 'ultrasound' in singular; line 50: better: 'low- to-moderate temperature'; change 'compounds structure' to 'structure of the compounds'; line 51: 'high pressure acts selectively since may disrupt' rephrase and add 'it'...and so on.
Answer: As suggested, the corrections were done.

Round 2
Reviewer 1 Report
Issues were almost addressed.
Reviewer 2 Report
The reviewer has no further annotations to make.